# Airborne observations reveal elevational gradient in tropical forest isoprene emissions

Dasa Gu[1,2], Alex B. Guenther[1,2], John E. Shilling[2], Haofei Yu[2], Maoyi Huang[2], Chun Zhao[2,†], Qing Yang[2], Scot T. Martin[3], Paulo Artaxo[4], Saewung Kim[1], Roger Seco[1], Trissevgeni Stavrakou[5], Karla M. Longo[6], Julio Tóta[7], Rodrigo Augusto Ferreira de Souza[8], Oscar Vega[9], Ying Liu[2], Manish Shrivastava[2], Eliane G. Alves[10], Fernando C. Santos[6], Guoyong Leng[2] & Zhiyuan Hu[2]

Isoprene dominates global non-methane volatile organic compound emissions, and impacts tropospheric chemistry by influencing oxidants and aerosols. Isoprene emission rates vary over several orders of magnitude for different plants, and characterizing this immense biological chemodiversity is a challenge for estimating isoprene emission from tropical forests. Here we present the isoprene emission estimates from aircraft eddy covariance measurements over the Amazonian forest. We report isoprene emission rates that are three times higher than satellite top-down estimates and 35% higher than model predictions. The results reveal strong correlations between observed isoprene emission rates and terrain elevations, which are confirmed by similar correlations between satellite-derived isoprene emissions and terrain elevations. We propose that the elevational gradient in the Amazonian forest isoprene emission capacity is determined by plant species distributions and can substantially explain isoprene emission variability in tropical forests, and use a model to demonstrate the resulting impacts on regional air quality.

[1] Department of Earth System Science, University of California, Irvine, California 92697, USA. [2] Atmospheric Sciences & Global Change Division, Pacific Northwest National Laboratory, Richland, Washington 99354, USA. [3] Department of Earth and Planetary Sciences, School of Engineering and Applied Sciences, Harvard University, Cambridge, Massachusetts 02138, USA. [4] Instituto de Fisica, Universidade de São Paulo, 05508-900 São Paulo, Brazil. [5] Department of Atmospheric Composition, Royal Belgian Institute for Space Aeronomy, Avenue Cirbulaire 3, B-1180, Brussels, Belgium. [6] Earth System Science Center, National Institute for Space Research, São José dos Campos, 12227-010 São Paulo, Brazil. [7] Instituto de Engenharia e Geociencias, Universidade Federal do Oeste do Pará, 66075-900 Belem, Para, Brazil. [8] Escola Superior de Tecnologia, Universidade do Estado do Amazonas, 69050-020 Manaus, Amazonas, Brazil. [9] Centro de Química e Meio Ambiente, Instituto de Pesquisas Energéticas e Nucleares, 03178-200 São Paulo, Brazil. [10] Department of Climate and Environment, National Institute for Amazonian Research, 69067-375 Manaus, Amazonas, Brazil. † Present address: School of Earth and Space Sciences, University of Science and Technology of China, Hefei, Anhui 230026, China. Correspondence and requests for materials should be addressed to D.G. (email: dasag@uci.edu) or to A.B.G. (email: alex.guenther@uci.edu).

Terrestrial vegetation emits vast quantities of volatile organic compounds (VOCs) to the atmosphere[1–3], which influence oxidants and aerosols leading to complex feedbacks on air quality and climate[4–6]. Isoprene is a short-lived (minutes to hours) reactive VOC species, and the photo-oxidation of isoprene affects the oxidation capacity of the atmosphere and can contribute to the formation of ozone ($O_3$) and secondary organic aerosol[5–8]. The biogenic sources from terrestrial plant foliage contribute more than 90% of atmospheric isoprene[2]. The Amazonian forest has the richest assemblage and abundance of vegetation species on Earth. Recent studies suggest that $\sim$1.4% of the $\sim$16,000 tree species in the Amazon are hyperdominant and account for half of all the Amazonian trees[9], and only $\sim$1% of tree species are responsible for half of all carbon storage and productivity[10]. It is still not clear how many plant species can emit substantial quantities of isoprene, how these isoprene emitters are distributed across the Amazon basin, what is the magnitude of the emission, and how it varies seasonally. Satellite observations of isoprene oxidation products (for example, formaldehyde, glyoxal) have given an initial view of the global dynamic distribution of biogenic isoprene emission but there remains a need to parametrize and evaluate the estimations with regional measurements especially in the Amazon[11]. Driven by land cover distributions, vegetation emission factors (EFs) and environmental conditions, the Model of Emissions of Gases and Aerosols from Nature (MEGAN) can estimate emission fluxes of biogenic isoprene and other VOCs using simple mechanistic algorithms to account for the major known processes controlling biogenic emissions[2,12]. The model has estimated tropical trees to be responsible for 80% of global terpenoid emissions[12], but emissions derived from satellite observations suggest those values are overestimated[13].

Here we report isoprene emission fluxes estimated from airborne measurements around Manaus, Brazil (3°06'S, 60°01'W) in the central Amazon Basin during the Green Ocean Amazon (GoAmazon) 2014/15 campaign[14]. Fast response airborne proton transfer reaction-mass spectrometry (PTR-MS) measurements of isoprene mixing ratios provide an opportunity to estimate isoprene emissions from this tropical forest using wavelet-based eddy covariance (EC) techniques. The EC technique, which provides the most direct measurement of fluxes, has recently been implemented for airborne VOC emission

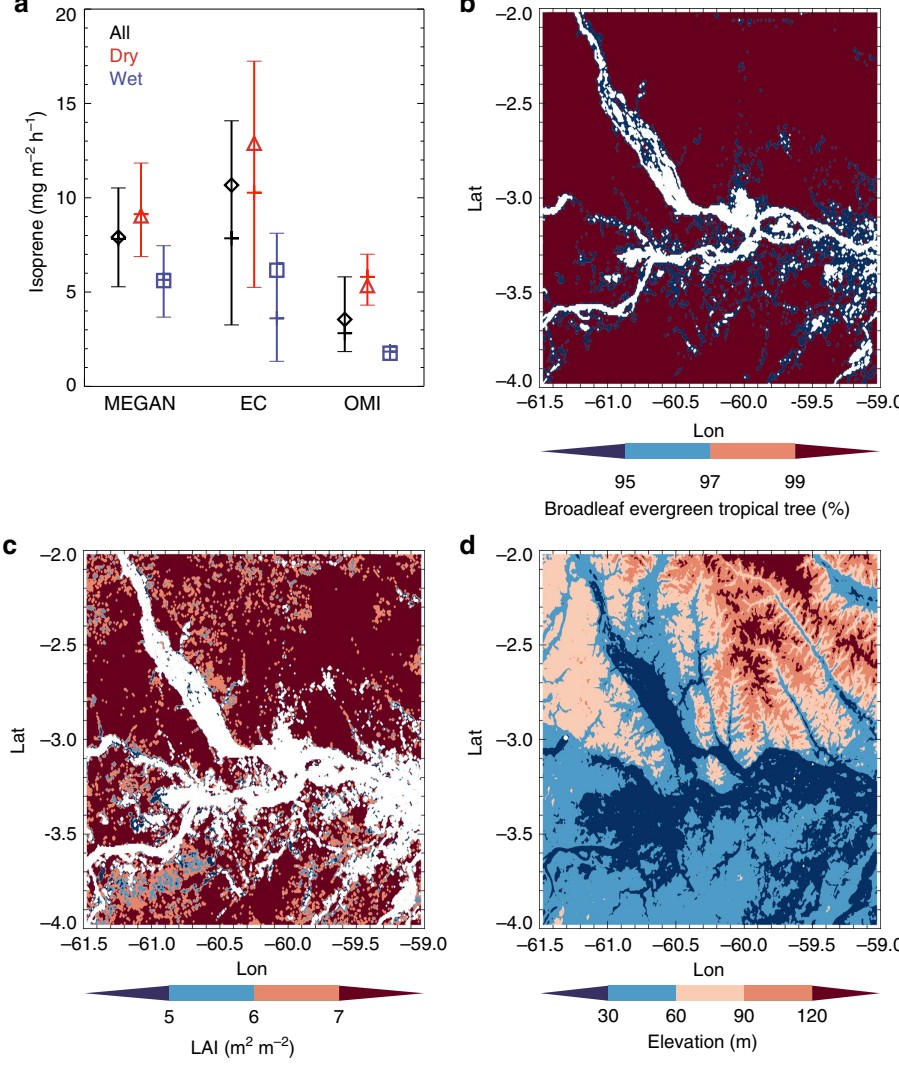

**Figure 1 | Isoprene emission estimates and maps of vegetation distributions and terrain elevation.** (**a**) Mean values of surface isoprene emissions from MEGAN, EC and OMI for all available flights (black diamond), dry season (red triangle) and wet season (blue square), and their 25% quartile values (lower bar), 50% quartile values (middle bar) and 75% quartile values (higher bar). (**b**) Fractional coverage of broadleaf evergreen tropical trees from MODIS PFT land cover observation. (**c**) Distribution of LAI in September 2014 from MODIS observation. (**d**) Terrain elevation from ASTER Global Digital Elevation Map.

measurements[15,16]. Eight research flights (RFs) in wet (January to June) and dry (July to December) seasons are selected for investigation based on flight maneuvers and environmental factors to minimize the impacts from city plumes (Supplementary Fig. 1). Utilizing the direct airborne measurements, this work proposes to: (1) elucidate the spatial heterogeneity of isoprene emission estimates from direct aircraft measurements over the Amazonian forest compared with model predictions based on satellite land cover and vegetation specific EFs; (2) quantify the elevational gradient in the Amazonian forest isoprene emission capacity with airborne observations and satellite top-down estimates; (3) use a regional chemical transport model to assess the impacts from the proposed changes in isoprene EFs on the major radicals and air pollutants. We observe isoprene emission rates that are three times higher than satellite top-down estimates and 35% higher than model predictions based on satellite land cover and vegetation specific EFs. The results reveal strong correlations between observed isoprene emission rates and terrain elevations which are confirmed by similar correlations between satellite-derived isoprene emissions and terrain elevations. By updating the isoprene EFs based on the observed magnitude and the relation between isoprene emissions and terrain elevations, there are significant impacts on regional oxidants distributions predicted by a regional model simulations.

## Results

**Observed and model simulated isoprene emission rates.** The average surface isoprene emission rates are 6.2, 12.9 and 10.7 mg m$^{-2}$ h$^{-1}$ from observations in wet, dry and both seasons based on the EC technique (Fig. 1a). The observed isoprene emission rates are about 3.5, 2.4 and 3 times higher than the estimates from a satellite top-down approach based on the Ozone Monitoring Instrument (OMI) measurements in wet, dry and both seasons. Compared with the estimates from MEGAN model simulations, the observed isoprene emission rates are 10%, 43% and 35% higher in wet, dry and both seasons. We also estimated emission estimates from the aircraft observations using an independent approach, the Mixed Layer Variance technique[17,18] (Supplementary Fig. 2), and the calculated isoprene emission rates are comparable with the direct EC measurements and estimates from previous studies[17,19].

To accurately simulate the spatiotemporal distribution of isoprene emissions with MEGAN, it is critical to drive the model with representative land cover input data including EFs, plant functional type (PFT) and leaf area index (LAI). In this study, we use the MEGAN v2.1 model coupled with the Community Land Model (CLM) v4.5 to simulate isoprene emissions over the central Amazon forest in 2014. The MEGAN v2.1 model adopts the 16 PFT scheme used by CLM to characterize spatial variations of vegetation types, and specifies isoprene EFs based on PFT categories[12]. Satellite observations have been widely used to generate high-resolution land cover parameters for Earth system modelling[20]. By using Moderate Resolution Imaging Spectroradiometer (MODIS) satellite data, we calculated MEGAN PFT and LAI inputs for this study (Fig. 1b,c). Based on the MODIS MCD12Q1 land cover type product, the study region is dominated by only one PFT type, broadleaf evergreen tropical trees, which results in a nearly homogeneous distribution of EFs for model simulations. Although there are small percentages of other PFT (for example, grass, crop) and water (river) coverage dispersed throughout the region, their estimated contributions to the overall isoprene emission are very small (Supplementary Fig. 3). Comparing LAI data from the MODIS MCD15A2 product in each month in 2014 (Supplementary Fig. 4), the LAI in the dry season is significantly higher than in the wet season, which contributes to higher isoprene emissions in the dry season. The MEGAN model is also driven by meteorological inputs (for example, temperature, radiation) from simulations of meteorological forcing data from the Weather Research and Forecasting (WRF) Model constrained by National Center for Environmental Prediction FiNaL (NCEP FNL) operational global analysis data. Vegetation temperature and solar radiation in the dry season were higher than those in the wet season (Supplementary Figs 5 and 6), which both tend to contribute to higher isoprene emissions in the dry season. The 24-hour monthly average isoprene emission is 4.3 mg m$^{-2}$ h$^{-1}$ in September from MEGAN simulation, which is nearly twice the emission of 2.1 mg m$^{-2}$ h$^{-1}$ in March (Supplementary Fig. 7).

By comparing the emissions estimated from airborne observations with those from MEGAN simulations, we evaluated the average MEGAN emission and propose an approach for improving the model estimates. As shown in Fig. 2, there are discrepancies in the spatial distributions of isoprene emissions between MEGAN model simulations and those derived from

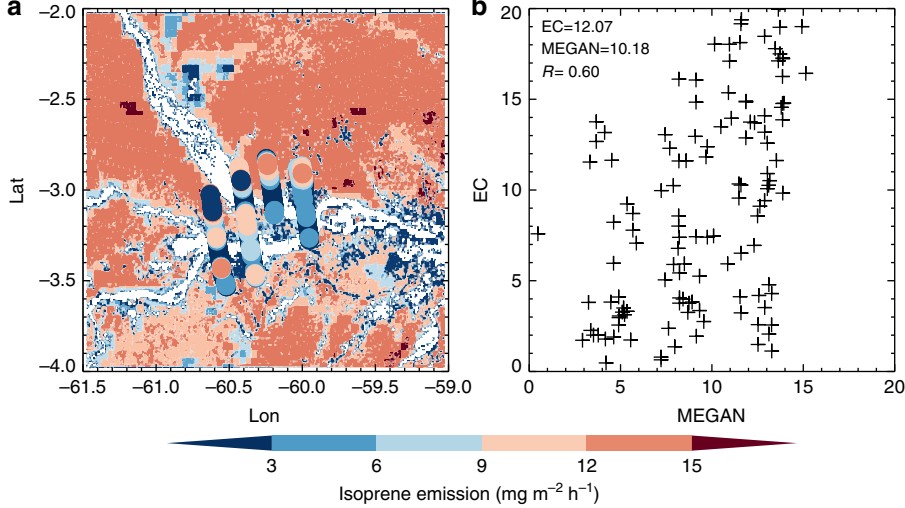

**Figure 2 | Surface isoprene emission flux during flight RF 20140930.** (**a**) Spatial distributions from airborne EC method (solid circles) compared with MEGAN simulations (background colours); (**b**) scatter plot of the EC and MEGAN estimates, and their mean values and linear correlation coefficient are shown in the figure..

aircraft observations. Also shown in Supplementary Fig. 8, the observed isoprene emission rates are 35% higher than average model results, while the emissions from aircraft observations are more variable indicating isoprene emission heterogeneity that is not captured by the model.

**Elevational gradient of isoprene emission.** To exclude the impacts from meteorological inputs, we calculated the isoprene EFs from aircraft observations, and compared them with corresponding MEGANv2.1 EFs. While the EFs in MEGANv2.1 are dominated by one single–PFT-based MODIS land cover data, the aircraft observed EFs are significantly more variable (Supplementary Table 1), suggesting that there is greater heterogeneity in actual vegetation types and isoprene emissions. While the Amazonian forest has the richest abundance of vegetation species on Earth, there remains much unknown about the plant species distribution in the Amazon[21]. The emission rate variability in this diverse ecosystem must be characterized by more than one single PFT to adequately represent the entire Amazon forest. While LAI can influence isoprene emission because it represents the magnitude of the potential source, there were no clear correlations between observed EF and LAI (Supplementary Table 2). Therefore, we investigated other variations in land characteristics that could explain this variability.

Variations in ecosystem types, and their associated plant species distributions, have been observed along elevational gradients in many regions and are associated with altitude driven changes in a variety of environmental factors (for example, temperature, humidity, soil composition)[22]. Studies have shown floristic compositions of tree[23], shrub[24] and palm[25] are correlated with terrain elevations in Amazonian forests. Therefore, we compared aircraft-based isoprene EF with satellite-based elevation data to investigate whether there are elevational gradients in isoprene EFs. As shown in Fig. 3, by categorizing the observed isoprene EFs using 30 m intervals in terrain elevations, there are strong positive correlations ($R = 0.98$, $P < 0.018$) between isoprene EFs and elevations for the dry season. This indicates that there is a notable elevational gradient of isoprene emitters in the central Amazonian forest. We hypothesize that an elevational gradient in Amazonian forest isoprene emission capacity, determined by plant species distributions, can explain a substantial degree of isoprene

emission variability in Amazonian tropical forests leading to significantly improved isoprene emission estimates.

We also examined biogenic isoprene emissions from top-down estimations based on the Global Ozone Monitoring Experiment–2 (GOME-2) (2007–2012) and OMI (2005–2014) satellite formaldehyde observations (Fig. 3). Similar to the EFs from aircraft observations, there are also strong correlations ($R = 0.96$–0.99) between top-down isoprene emissions and terrain elevation in the central Amazon. The top-down emissions are impacted by the a priori emission (MEGAN-MOHYCAN[26]) which has lower values at lower elevations due to the combination of river, grassland with trees in the low resolution (0.5 degree) grid. As a result, the elevational variation of vegetation composition could be impacted by the assumed land cover types. Based on aircraft observed isoprene EF and terrain elevation data, the observed relationship (EF = 0.091 × Elevation + 4.51) in dry season was used to modify the isoprene EFs in the central Amazon. As shown in Fig. 4, the revised EFs are consistently higher than the MEGANv2.1 EFs and there is significant horizontal heterogeneity of EFs with higher values in the northern part of the study domain. On average, the revised EFs are 71% higher than the MEGANv2.1 EFs in the study domain.

**Model simulated regional impacts.** To examine the impacts of the revised isoprene emissions, we used the Weather Research and Forecasting model coupled to Chemistry (WRF-Chem), to simulate the impact of the improved isoprene EFs on regional oxidants distributions as shown in Fig. 5. The hydroxyl radical (OH), which is the primary oxidant for most tropospheric trace gases (for example, nitrogen oxides ($NO_x$), formaldehyde (HCHO)), decreased by ∼19% after updating the isoprene EFs. At the same time, two important photochemical oxidation products, $O_3$ and peroxyacyl nitrates, decreased by ∼10% and ∼6%. This suggests that the higher isoprene emission is currently suppressing these compounds since the $O_3$–$NO_x$-VOC sensitivity is $NO_x$-limited in the Amazonian area. This will likely not be the case if $NO_x$ emissions increase as a result of increased anthropogenic activities in the Amazon[27]. As the elevational gradient of the plant species distribution is also related to the height above the nearest drainage in the Amazonian forest[28], we may also see future changes of biogenic isoprene emission and regional air quality if the water table depth fluctuates as a consequence of climate change or human activities.

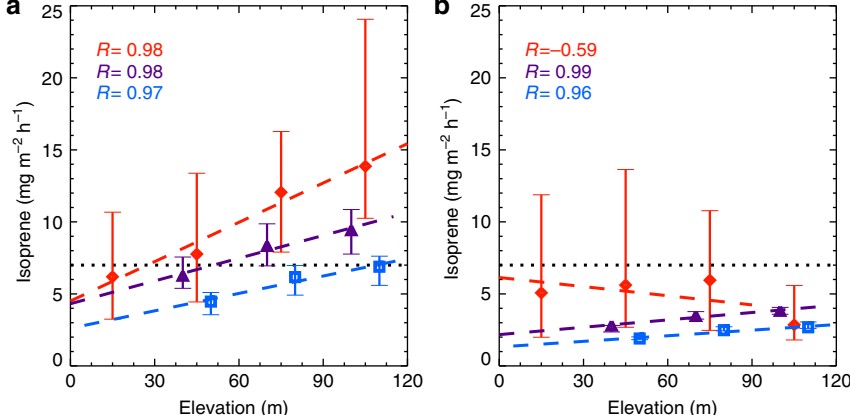

**Figure 3 | Correlations of terrain elevations with observed isoprene EFs and top-down isoprene emissions.** The median values of isoprene EFs estimated from EC approach (red diamond), top-down biogenic isoprene emissions based on satellite data including GOME-2 (purple triangle) and OMI (blue square), and their 25% quartile values (lower bar) and 75% quartile values (higher bar) during dry (**a**) and wet (**b**) seasons compared with terrain elevations with an interval of 30 m. The black dot lines indicate the EF used in MEGAN v2.1. The colour dash lines show linear regressions for median values from each approach, and their correlation coefficients (R) are shown in the figures.

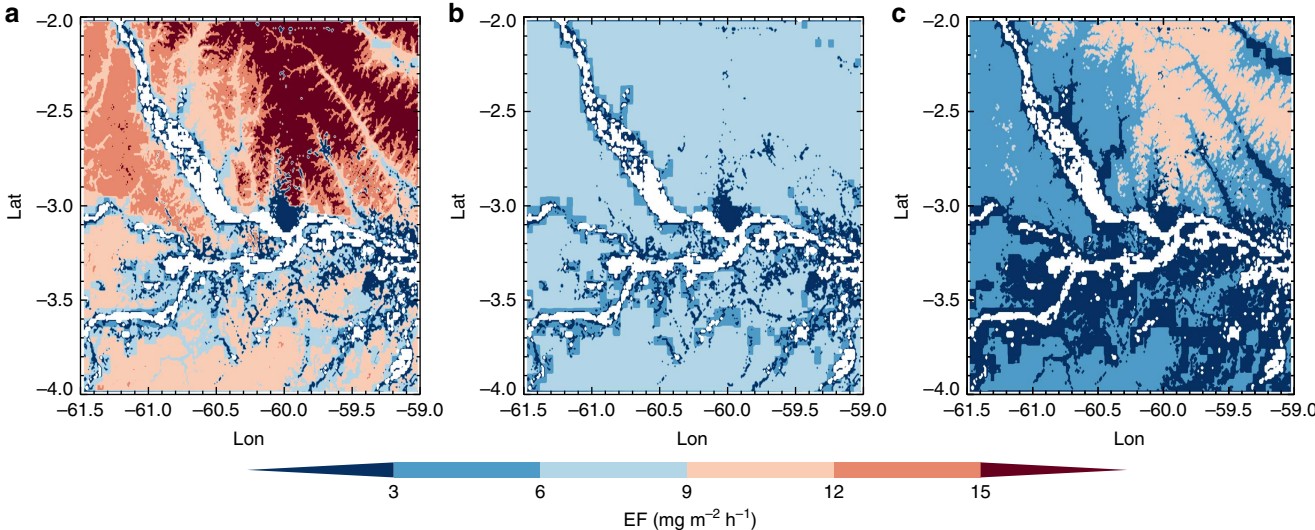

**Figure 4 | Distribution of isoprene EF.** Comparison of isoprene EFs based on observations from airborne EC approach (**a**), based on MEGANv2.1 EFs and MODIS PFT land cover observations (**b**) and the difference between the above two data sets (**c**).

**Figure 5 | Impact of revised isoprene EF.** Relative changes of mixing ratios of $O_3$ (**a**), $NO_x$ (**b**), isoprene (**c**), OH (**d**), peroxyacyl nitrates (PAN) (**e**) and HCHO (**f**) in the boundary layer by updating the isoprene EFs simulated by WRF-Chem during September 2014; the mean value of the whole domain is reported in parentheses.

## Discussion

Our study provides the first evidence of elevational gradients of isoprene emissions in the Amazonian forest, indicates significantly higher isoprene EFs compared with previous estimates, and demonstrates the important implications for regional atmospheric photochemistry and air quality. Furthermore, these observations show that biogenic isoprene emissions are much higher in the dry season than in the wet season, and the isoprene EFs are higher than those assumed for previous MEGAN simulations. The revised isoprene EFs based on aircraft observations and terrain elevation distributions account for the notable heterogeneity of isoprene emission in the central Amazonian forest, and lead to significant impacts on photochemistry and regional air quality. This study provides an important initial demonstration of the significance of elevational gradients in biogenic isoprene emissions in the Amazon tropical rain forest. Further measurements of leaf and canopy-scale isoprene emissions at multiple sites along elevation gradients, particularly over broader tropical regions, are needed to determine the cause and the generality of the relationship in other geographic regions.

## Methods

**Airborne measurements.** Isoprene was measured by a PTR-MS onboard the Gulfstream-1 (G-1) research aircraft around Manaus, Brazil during the GoAmazon2014/5 campaign in both wet and dry seasons 2014. A detailed description of the PTR-MS and other gas species (for example, carbon monoxide (CO), $O_3$ and $NO_x$) measurements on the G-1 aircraft is provided in ref. 29. The observations from four wet season flights and four dry season flights used in this study are shown in Supplementary Note 1.

**EC techniques.** The EC technique based on wavelet analysis estimates the turbulent flux $F$ as the discrete covariance between the fluctuating terms of vertical wind speed ($w'$) and concentrations ($C'$):

$$F = \sum w' \times C' \qquad (1)$$

Using wavelet transformation, the EC technique computes instantaneous correlations between $w'$ and $C'$ (that is, isoprene) to get flux estimates with high spatial resolution ($\sim 2\,km$). Afterwards the surface emission flux is derived from the high-resolution flux data of EC together with a vertical flux divergence correction. A detailed description of the airborne EC technique is provided in refs 15,16.

**MEGAN simulation within CLM 4.5 framework.** MEGAN is a global biogenic emission model that is used for both regional air quality modelling and global climate and Earth system modelling studies and is driven by meteorology and land cover data[12]. The MEGAN model has been embedded into land surface, chemical transport and global climate models. In this study, we utilized MEGAN v2.1 integrated into CLM 4.5 with a resolution of $\sim 1\,km$ for the study domain of 4°S to 2°S and 61.5°W to 59°W. The land cover inputs (for example, PFT, LAI) are derived from satellite observations. The model ran for the entire year of 2014 using meteorological forcing data derived from WRF simulations (constrained by NCEP FNL data).

**Satellite observations.** MODIS satellite product MCD12Q1 land cover type product with 500 m resolution was aggregated to 1 km grids to derive PFT distributions over the model domain. We used the latest MCD12Q1 data set for 2012 in this study. MODIS MCD15A2 LAI 8-day composite product with 1 km resolution was used in this study for the entire year of 2014. The PFT and LAI mapping into CLM classification follows ref. 20. The ASTER Global Digital Elevation Model (ASTER GDEM) v2 in $\sim 30\,m$ resolution was used to calculate the terrain elevation for this study (Fig. 1d). The top-down biogenic isoprene emissions source inversion using the adjoint of the IMAGESv2 global chemistry-transport model[11] was constrained by tropospheric HCHO column densities from the OMI and GOME-2 satellite instrument[30].

**WRF-Chem simulation.** The WRF-Chem (v3.5.1) configuration is described in ref. 31. The SAPRC-99 (Statewide Air Pollution Research Center 1999) photochemical mechanism was selected to simulate gas-phase chemistry, and the Fast-J parameterization for photolysis rates[31]. The model simulation is between 6 and 30 September 2014, when most of the dry season measurements were conducted.

**Code availability.** The CLM 4.5 model code used in this study is available at http://www.cesm.ucar.edu/models/cesm1.2/clm/. The WRF-Chem model code used in this study can be found at http://ruc.noaa.gov/wrf/wrf-chem/.

**Data availability.** All of the airborne measurement data used to calculate the isoprene emissions are available at http://www.arm.gov/campaigns/amf2014goamazon. The MODIS PFT and LAI data can be found at https://lpdaac.usgs.gov/dataset_discovery/modis/modis_products_table. The ASTER Global Digital Elevation Map data is available at https://asterweb.jpl.nasa.gov/gdem.asp. The top-down biogenic isoprene emission data is available at http://www.globemission.eu/data.php.

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

## Acknowledgements

Institutional support was provided by the Central Office of the Large Scale Biosphere Atmosphere Experiment in Amazonia (LBA), the National Institute of Amazonian Research (INPA), the National Institute for Space Research (INPE), Amazonas State University (UEA/FAPEAM) and the Brazilian Space Agency (AEB). The work was conducted under 001262/2012-2 of the Brazilian National Council for Scientific and Technological Development (CNPq). We acknowledge the Atmospheric Radiation Measurement (ARM) Climate Research Facility, a user facility of the United States Department of Energy, Office of Science, sponsored by the Office of Biological and Environmental Research, and support from the Atmospheric System Research (ASR) programme of that office. A.B.G. was partially supported by the National Aeronautics and Space Administration (NASA) Atmospheric Composition Campaign Data Analysis and Modeling (ACCDAM) programme award NNX15AT62G. We thank Juliana Schietti for discussions.

## Author contributions

S.T.M., A.B.G., P.A., K.M.L., J.T., R.A.F.d.S. and O.V. planned and organized the project. J.E.S. conducted measurements of isoprene by PTR-MS. D.G., A.B.G., J.E.S., H.Y. and Q.Y. analysed airborne measurement data. D.G. and A.B.G. analysed and modelled the data, and wrote the manuscript. D.G., M.H., C.Z., Y.L., G.L., Z.H. and F.C.S. conducted CLM simulations. C.Z., Z.H. and M.S. conducted WRF-Chem simulations. T.S. derived biogenic isoprene emission from satellite observations. J.E.S., M.H., C.Z., Q.Y., S.T.M., S.K., R.S., T.S., J.T., E.G.A. and M.S. revised the manuscript. All authors reviewed and commented on the paper.

## Additional information

**Competing interests:** The authors declare no competing financial interests.

