## [Peer Review File · Nature Communications]

Reviewers' Comments:

Reviewer #1 (Remarks to the Author)

Gu et al. report on studies of isoprene and other gases in the atmosphere over the central Amazon Basin. Isoprene plays a major role in atmospheric chemistry globally but especially in the Amazon Basin. Although it is well known that the total flux of isoprene to the atmosphere is massive, precise estimates are difficult because not all plants make isoprene and the effects of dry versus wet season is hard to predict. Large scale measurements available today rely either on measuring leaves and scaling up or looking down from satellites. This report provides data from aircraft measurements using eddy correlation to measure flux at the landscape scale. This is the most precise method for making measurements at this scale. The measurements will be most helpful in parameterizing global models. What is more, the authors found an effect I have not heard of before. They found that elevation has a significant effect on isoprene emission. It will be interesting in the coming years to find out the physiological reason for this unanticipated observation.

The paper is well written and is accessible to both the specialist and non specialist. The novel findings are the altitude effect and that isoprene emissions are higher than current models predict. This is especially important because there has been a debate whether increasing CO₂ in the atmosphere will inhibit isoprene emission. Now that CO₂ is permanently above 400 ppm, finding higher than expected isoprene flux indicates that there is no evidence yet for a CO₂ suppression of isoprene emission.

The manuscript is remarkably free of typos. I have two minor suggestions regarding the figures: 1. In figure 1 (and others) it took me a second to realize the axes are latitude and longitude in degrees. This should be in the figure legend if not actually on the graphs themselves. 2. In figures 1 and 3 the same symbol is used but different colors to distinguish among data. I suggest using different symbols as well as color to distinguish data.

I anticipate this paper will receive wide attention.

Tom Sharkey

Reviewer #2 (Remarks to the Author)

The paper presents eddy covariance measurements of isoprene from an aircraft. Comparison of fluxes measured using the EC technique and those calculated from the MEGAN model show a discrepancy of over 30%. Given the assumptions made by the model these differences are not surprising. The discrepancies between model and measurements was shown to be accountable for by vegetation variation with altitude, which is not taken into account by MEGAN across the varied topography of the Amazon..

These EC measurement fluxes will help to improve the parameters used within such models. Using emission factors for wet and dry seasons and a more flexible plant functional type and leaf area index will help improve model estimates of biogenic VOC fluxes.

The Morlet wavelet transformation is used to calculate eddy covariance flux estimated from the flight data collected. The paper and data forms a convincing case for the conclusion drawn. The one criticism would be in the data sample numbers used for some of the calculation parameters (see tables S1 & 2).

Reviewer #3 (Remarks to the Author)

The authors report measured emissions from the Amazon with a relatively new technology and found that these are higher than previous estimates from models and satellite information. They argue conclusively that the reason for this divergence is the heterogeneity of the vegetation that is generally treated as homogeneous but actually is not. Finally they suggest a mechanism for stratification that improves emission estimates and is relatively easy to apply.

The manuscript is well written, covers an interesting subject where it adds provides novel information and should be influential for emission modelling world-wide. I cannot find major flaws that may prevent publication in the information given.

Some very minor points are the following

It is a bit confusing that tropical trees are stated as responsible for 80 percent of global isoprene emissions (L54) as well as 80 percent of global terpenoids emissions (L70), which appears to imply that the same share of global isoprene and monoterpenes are emitted from tropical trees. This, however, is not the case given the larger share of monoterpene emissions by boreal forests but is due to different literature references.

It should be noted somewhere that the isoprene emissions derived from EC measurements have a tendency to underestimate the flux because isoprene is also deposited or chemically destroyed before reaching the aircraft. On the other hand, it may be noteworthy that the PTR-MS procedure usually is not able to differentiate isoprene from a couple of other (minor) compounds.

Please give a reference to the MEGAN-Mohycan model (perhaps Mueller et al. 2008?) and perhaps indicate that this is a multilayer approach.

L224-26 are basically repeating the sentence before. This is probably a mistake.

I am not sure if this has been indicated somewhere but I would like to read about the temporal resolution of images where the LAI dynamics are derived from.

Response to referees' comments

Reviewer #1 (Remarks to the Author):

Gu et al. report on studies of isoprene and other gases in the atmosphere over the central Amazon Basin. Isoprene plays a major role in atmospheric chemistry globally but especially in the Amazon Basin. Although it is well known that the total flux of isoprene to the atmosphere is massive, precise estimates are difficult because not all plants make isoprene and the effects of dry versus wet season is hard to predict. Large scale measurements available today rely either on measuring leaves and scaling up or looking down from satellites. This report provides data from aircraft measurements using eddy correlation to measure flux at the landscape scale. This is the most precise method for making measurements at this scale. The measurements will be most helpful in parameterizing global models. What is more, the authors found an effect I have not heard of before. They found that elevation has a significant effect on isoprene emission. It will be interesting in the coming years to find out the physiological reason for this unanticipated observation.

The paper is well written and is accessible to both the specialist and non specialist. The novel findings are the altitude effect and that isoprene emissions are higher than current models predict. This is especially important because there has been a debate whether increasing CO₂ in the atmosphere will inhibit isoprene emission. Now that CO₂ is permanently above 400 ppm, finding higher than expected isoprene flux indicates that there is no evidence yet for a CO₂ suppression of isoprene emission.

We thank the reviewer for the constructive comments and great support for publishing our study! We have revised the manuscript following the advice.

The manuscript is remarkably free of typos. I have two minor suggestions regarding the figures: 1. In figure 1 (and others) it took me a second to realize the axes are latitude and longitude in degrees. This should be in the figure legend if not actually on the graphs themselves. 2. In figures 1 and 3 the same symbol is used but different colors to distinguish among data. I suggest using different symbols as well as color to distinguish data.

As suggested by the reviewer, we have added the labels of latitude and longitude to all map style figures.

In Figure 1 and 3, we have changed to use different symbols as well as colors to distinguish different datasets.

I anticipate this paper will receive wide attention.

Tom Sharkey

Reviewer #2 (Remarks to the Author):

The paper presents eddy covariance measurements of isoprene from an aircraft. Comparison of fluxes measured using the EC technique and those calculated from the MEGAN model show a discrepancy of over 30%. Given the assumptions made by the model these differences are not surprising. The discrepancies between model and measurements was shown to be accountable for by vegetation variation with altitude, which is not taken into account by MEGAN across the varied topography of the Amazon..

These EC measurement fluxes will help to improve the parameters used within such models. Using emission factors for wet and dry seasons and a more flexible plant functional type and leaf area index will help improve model estimates of biogenic VOC fluxes.

The Morlet wavelet transformation is used to calculate eddy covariance flux estimated from the flight data collected. The paper and data forms a convincing case for the conclusion drawn. The one criticism would be in the data sample numbers used for some of the calculation parameters (see tables S1 & 2).

We thank the reviewer for the constructive comments and great support for publishing our study!

For Supplementary Table 1, it shows the isoprene EFs in MEGANv2.1 model are dominated by one single PFT (Broadleaf evergreen tropical trees). When we are comparing our airborne observation data with the MEGANv2.1 EFs, nearly all observation data (418 data points in wet season and 848 data points in dry season) are considered to be broadleaf evergreen tropical trees with isoprene EF of $7 \text{ mg m}^{-2} \text{ h}^{-1}$ in the models. However, the observations not only show higher EFs, but also show significant heterogeneous distributions (Figure 2). It is worth noting that the EC EFs in the table are directly observed EFs that have not been adjusted for the PFT coverage fractions in the model grids.

For Supplementary Table 2, it shows our effort to explain the correlations between the observed isoprene EFs and terrain elevations. While LAI is expected to significantly influence the vegetation isoprene emissions because it is related to the magnitude of the potential source, we investigated whether LAI could explain the variations in observed isoprene EFs. However, there are no clear correlations between observed EF and LAI, which encouraged us to investigate other potential parameters, such as terrain elevation.

Reviewer #3 (Remarks to the Author):

The authors report measured emissions from the Amazon with a relatively new technology and found that these are higher than previous estimates from models and satellite information. They argue conclusively that the reason for this divergence is the heterogeneity of the vegetation that is generally treated as homogeneous but actually is not. Finally they suggest a mechanism for stratification that improves emission estimates and is relatively easy to apply.

The manuscript is well written, covers an interesting subject where it adds provides novel information and should be influential for emission modelling world-wide. I cannot find major flaws that may prevent publication in the information given.

We thank the reviewer for the constructive comments and great support for publishing our study! We have revised the manuscript following the advice.

Some very minor points are the following

It is a bit confusing that tropical trees are stated as responsible for 80 percent of global isoprene emissions (L54) as well as 80 percent of global terpenoids emissions (L70), which appears to imply that the same share of global isoprene and monoterpenes are emitted from tropical trees. This, however, is not the case given the larger share of monoterpene emissions by boreal forests but is due to different literature references.

We thank the reviewer for the correction. We have checked the references, and have corrected the statement at Line 54 now.

It should be noted somewhere that the isoprene emissions derived from EC measurements have a tendency to underestimate the flux because isoprene is also deposited or chemically destroyed before reaching the aircraft. On the other hand, it may be noteworthy that the PTR-MS procedure usually is not able to differentiate isoprene from a couple of other (minor) compounds.

In the method section on EC technique, it is described that ‘the surface emission flux is derived from the high-resolution flux data of EC together with a vertical flux divergence correction’. Following the methodology described at Karl et al., [2009] and Misztal et al. [2014], we utilized the measurements at multiple attitudes to interpolate the vertical flux divergence between the top of the canopy and the height of the aircraft measurements.

We thank the reviewer for the question of PTR-MS procedure about isoprene measurements. We have added more details in the Supplementary Note 1: ‘We assume that the PTR-MS signal at m/z 69 is only isoprene but it should be noted that there are potential interferences for this mass including fragmentation of large molecule species in diesel or gasoline exhaust, furan from biomass burning, and alkene species. These impacts are expected to be minor for the relatively clean environment of the pristine Amazon forest, and we have minimized the impacts from city plumes and biomass burning by excluding the flight segments with high values of O₃, CO, NO_x and aromatics (as indicators of biomass

burning and anthropogenic pollutants).'

Please give a reference to the MEGAN-Mohykan model (perhaps Mueller et al. 2008?) and perhaps indicate that this is a multilayer approach.

We have added the reference in the manuscript.

L224-26 are basically repeating the sentence before. This is probably a mistake.

Thanks! We have revised the sentences.

I am not sure if this has been indicated somewhere but I would like to read about the temporal resolution of images where the LAI dynamics are derived from.

In the method section on Satellite observations, it is described that 'MODIS MCD15A2 LAI 8-day composite product with 1 km resolution was used in this study for the entire year of 2014'.